# Using a Digital Microfluidic System to Evaluate the Stretch Length of a Droplet with a L-DEP and Varied Parameters

**Hsiang-Ting Lee [1], Ying-Jhen Ciou [1] and Da-Jeng Yao [1,2,*]**

[1]   Department of Power Engineering, National Tsing Hua University, Hsinchu 30013, Taiwan;
     linkingpark2655@yahoo.com.tw (H.-T.L.); janetciou1022@gmail.com (Y.-J.C.)
[2]   Institute of NanoEngineering and MicroSystems, National Tsing Hua University, Hsinchu 30013, Taiwan
*    Correspondence: djyao@mx.nthu.edu.tw; Tel.: +886-3-571-5131 (ext. 42850); Fax: +3-574-5454

**Abstract:** Digital microfluidics has become intensively explored as an effective method for liquid handling in lab-on-a-chip (LOC) systems. Liquid dielectrophoresis (L-DEP) has many advantages and exciting prospects in driving droplets. To fully realize the potential benefits of this technique, one must know the droplet volume accurately for its distribution and manipulation. Here we present an investigation of the tensile length of a droplet subjected to a L-DEP force with varied parameters to achieve precise control of the volume of a droplet. Liquid propylene carbonate served as a driving liquid in the L-DEP experiment. The chip was divided into two parts: an electrode of width fixed at 0.1 mm and a total width fixed at 1 mm. Each had a variation of six electrode spacings. The experimental results showed that the stretching length decreased with decreasing electrode width, but the stretching length did not vary with an increased spacing of the electrode. When the two electrodes were activated, the length decreased because of an increase in electrode spacing. The theory was based on the force balance on a droplet that involved the force generated by the electric field, friction force, and capillary force. The theory was improved according to the experimental results. To verify the theoretical improvement through the results, we designed a three-electrode chip for experiments. The results proved that the theory is consistent with the results of the experiments, so that the length of a droplet stretched with L-DEP and its volume can be calculated.

**Keywords:** L-DEP; digital microfluidics system

## 1. Introduction

Microfluidic systems in design and research are used to control and manipulate liquids in small amounts, from microliters to picoliters. Microfluidics have become greatly significant in chemical and biological operations and measurements [1–3], because of their many advantages, which include decreased volume, modest consumption of energy, brief duration of analysis and multiple functions in the same device [4]. Methods to control and to manipulate small amounts of liquid in microfluidic devices include digital microfluidics [5–7], droplet microfluidics and continuous flow microfluidics [8,9].

The lab-on-a-chip (LOC) technique uses a digital microfluidic system to transport and to manipulate a small amount of liquid. Electromagnetically driven microfluidic flow in digital microfluidic systems, include electro-osmosis [10], electroconvection [11], electrocapillary [12], electrowetting on dielectric (EWOD) [13,14] and liquid dielectrophoresis (L-DEP) [15,16]. Among these microfluidic mechanisms, EWOD and L-DEP have received much attention because of their low cost, modest consumption of power and avoidance of mechanical components such as pumps and valves. These two main mechanisms generate micro droplets and then transport, mix and manipulate them. For chips for

biological transport and analysis, a droplet might contain a desired particulate matter, such as cells, DNA and particles [17,18].

The difference between the mechanisms of L-DEP and EWOD actuation is the actuation voltage and frequency. In the EWOD system, droplets are placed between two parallel plates and actuated under a difference in wettability between the electrodes. EWOD occurs at low frequency and voltages; the applied voltage mostly distributes the dielectric layer [19]. The driving liquid is a conductive liquid. In the L-DEP system, droplets are placed on coplanar electrodes. When a voltage is applied, the liquid becomes polarized and flows to a region of stronger electric field. L-DEP requires a higher voltage and frequency than EWOD, and the driving liquid is a dielectric liquid [20].

To realize the potential benefit of L-DEP, one must accurately dispense and manipulate droplets of known volume in a rapid and controllable manner. Ahmed et al. improved L-DEP dispensing of a droplet by optimizing the electrode structures and analytical approaches [21]. Prakash et al. used L-DEP and tapered electrode structures to demonstrate that SMF devices can drive and distribute a range of droplets of varied volume in a reliable manner [22].

In this work we propose a theory for L-DEP based on parallel electrodes. We studied, theoretically and experimentally, the stretch length of a droplet subjected to an L-DEP force under applied voltage, electrode width and electrode spacing parameters. According to this theory, with a known electrode width and gap height, we can precisely control the droplet volume in a high-voltage and high-frequency environment.

## 2. Theory

In this section, we explore the forces on a droplet in an EWOD environment. The forces are divided mainly into a driving force on applying a voltage to generate an electric field to advance the droplet, the resistance of the capillarity force and the viscous force and the friction force on the contact line. Using this force balance to design a set of theories, we calculate the stretch length of the droplet.

When a droplet is located on an L-DEP chip, a voltage is applied to the electrodes that serve as upper and lower plates. The electric field is hence generated as Figure 1a shows. The experiment assumes that the spacing between the electrodes is much larger than the thickness of the dielectric layer, to investigate the distribution of the electric field in the droplets and the air, respectively $E_\mathrm{d}$, $E'_\mathrm{d}$, $E_\mathrm{l}$ and $E_\mathrm{w}$ [22]. Using an equivalent-circuit model, and ignoring the double-layer capacitance at the liquid-gas interface, because it is much larger than the capacitance of the dielectric layer [23–25], as shown in Figure 1b, the capacitances are respectively expressed as,

$$c_d = \kappa_d \epsilon_0 / d. \tag{1}$$

$$c_1 = \kappa_1 \epsilon_0 / D \tag{2}$$

$$c_w = \kappa_w \epsilon_0 / D \tag{3}$$

$$g_w = \sigma_w / D \tag{4}$$

in which $\kappa_d$ is the relative permittivity of the dielectric layer, $\kappa_1$ is the relative permittivity of a droplet, $\epsilon_0$ is the permittivity of vacuum, $d$ is the thickness of SU8 (epoxy-based negative photoresist) and $\sigma_w$ is the electrical conductivity. Substitution yields these formulae,

$$E = \frac{V}{D} \tag{5}$$

$$V(t) = \mathrm{Re}\left[\left(\sqrt{2}\right)V e^{j\omega t}\right] \tag{6}$$

in which $\omega$ is the angular frequency, $V$ is the rms voltage and $D$ is the spacing between the electrodes. Hence, we obtain the values of the electric field:

$$E_d = \frac{c_1}{2c_1 + c_d} V/d \tag{7}$$

$$E_l = \frac{c_d}{2c_1 + c_d} V/D \tag{8}$$

$$E'_d = Re\left[\frac{j\omega c_w + g_w}{j\omega(2c_w + c_d) + 2g_w} V/d\right] \tag{9}$$

$$E_w = Re\left[\frac{j\omega c_d}{j\omega(2c_w + c_d) + 2g_w} V/D\right] \tag{10}$$

According to Maxwell's stress tensor, as shown in Equation (11), in the closed case, the sum of the forces generated by the object in the z direction is

$$T^e_{mn} = \epsilon E_m E_n - \delta_{mn}\frac{1}{2}[\epsilon - (\partial\epsilon/\partial\rho)\rho]E_k E_k \tag{11}$$

$$F^e_z = \oint_{\Sigma} T^e_{mn} n_n dA \tag{12}$$

in which $n_n$ is the unit normal on the $n$th face of $\Sigma$; the z direction is assumed to be the direction in which the droplet moves. From Equations (7)–(12), the time-averaged force per unit length of the contact line generated by the electric field is

$$F^e_z = -\kappa_d\epsilon_0 E_d{}^2 d - \frac{\kappa_l\epsilon_0 E_l{}^2}{2}D + \kappa_d\epsilon_0 E'_d{}^2 d + \frac{\kappa_w\epsilon_0 E_w{}^2}{2}D \tag{13}$$

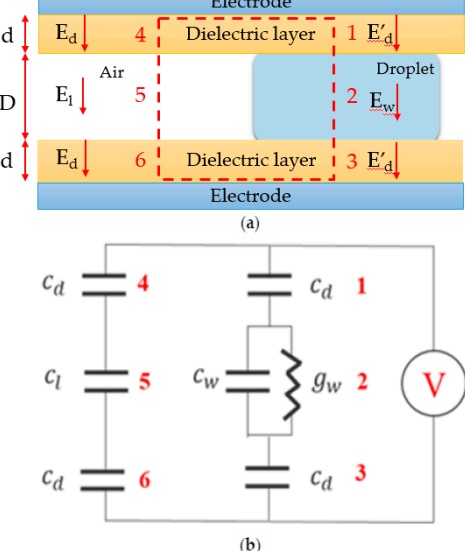

**Figure 1.** (**a**) Schematic diagram of a two-plate electrowetting on dielectric (EWOD) system; (**b**) equivalent-circuit model.

When a voltage is applied to the EWOD chip, a droplet is subjected mainly to four forces to achieve a force balance [26]. The first is the force generated by the electric field, and the other three forces are the viscous forces,

$$F_v = -\frac{4\mu}{c}z\frac{dz}{dt} \tag{14}$$

in which μ is the viscosity of the droplet, *c* is a dimensionless constant, and z is the stretch length of the droplet. The friction force at the contact line is proportional to the product of velocity and the transverse projected width of the contact line [27],

$$F_{cf} = -(2w + g)\xi\frac{dz}{dt}$$ (15)

in which ξ is the friction coefficient, *w* is the width of electrode, and *g* is the spacing between the electrodes. The capillarity force [28] is

$$F_{st} = -\pi\left(w + \frac{g}{2}\right)\gamma$$ (16)

in which γ is the surface tension of the liquid-gas interface and $\frac{dz}{dt}$ in the formula is assumed to be the average velocity. So $\frac{dz}{dt} = \frac{z}{t}$, the speed parameter becomes the average speed of the experiment. As the viscous force is about one-thousandth of the other two resistances after calculation, we neglected it in the calculation. The force balance of the relation between the electric field force, the friction force and the capillary force were eventually calculated, in Equations (3)–(27). Adding the friction force to the surface tension divided by the force generated by the electric field generates the stretch length of the droplet, as $F_z^e$ is the force per unit length. After bringing the experimental parameters into this formula and multiplying by the error coefficients, the results can be discussed and compared with the experiment.

$$F_z^e = -\kappa_d\epsilon_0 E_d{}^2\mathrm{d} - \frac{\kappa_l\epsilon_0 E_l{}^2}{2}D + \kappa_d\epsilon_0 E'{}_d{}^2 d + \frac{\kappa_w\epsilon_0 E_w{}^2}{2}D = (2w + g)\xi\frac{dz}{dt} + \pi\left(w + \frac{g}{2}\right)\gamma$$ (17)

## 3. Experiments

### 3.1. L-DEP Chip Design and Fabrication

The chip consists of a reservoir electrode, control electrodes and L-DEP electrodes as Figure 2a shows. The reservoir electrode has a 4 mm radius. Control electrodes in each pair have two sub-electrodes—an actuation electrode and a ground electrode. The control electrode serves as a channel connecting the reservoir and the L-DEP electrode, and is designed to be embedded in the reservoir to facilitate the generation of a droplet.

The L-DEP electrodes used in the experiment are of two types, one is called a two-electrode and the other a three-electrode. The two-electrode type is made up of two sets of rectangular electrodes, as Figure 2b shows; the length is fixed at 30 mm, the spacing between the electrodes is 20 μm. The total width is the width of the two electrodes plus the spacing between the electrodes, mainly divided into a total width of 1 mm and an electrode of width 0.1 mm. The label on the right side of the total width is the width of the electrode and the label on the right side of the electrode width is the size of the electrode spacing. The three-electrode type is made up of three sets of rectangular electrodes, as Figure 2c shows. The length is also fixed at 30 mm; the spacing between the electrodes is 20 μm. The total width of 1 mm is mainly divided into an electrode width 0.1 mm.

The fabrication of the L-DEP chip is shown in Figure 3. The L-DEP chip consisted of two parallel plates; the bottom substrate and the top plate were made of transparent indium tin oxide (ITO). The ITO glass was patterned with photolithography to create the bottom electrodes. The ITO electrodes on the bottom substrate were coated with a layer of SU8 photoresist (thickness 1.8 μm) to create a dielectric layer. The surfaces of the top plate and SU8 photoresist were coated with Teflon (1 μm) to create a hydrophobic layer.

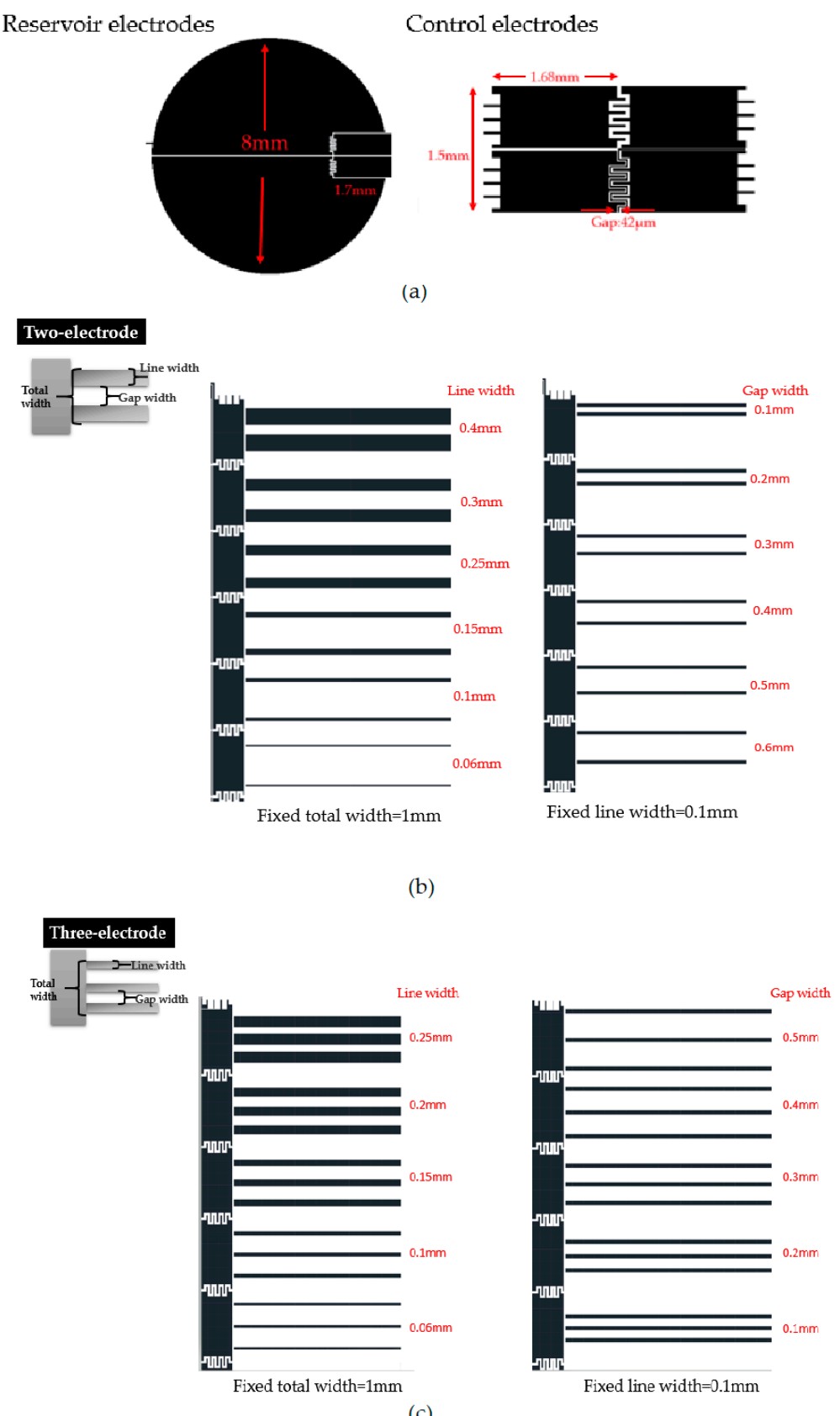

**Figure 2.** Schematic diagram of a parallel plate L-DEP chip (**a**) reservoir and control electrodes, (**b**) L-DEP two-electrode design with fixed total width and fixed line width and (**c**) L-DEP three-electrode design with fixed total width and fixed line width.

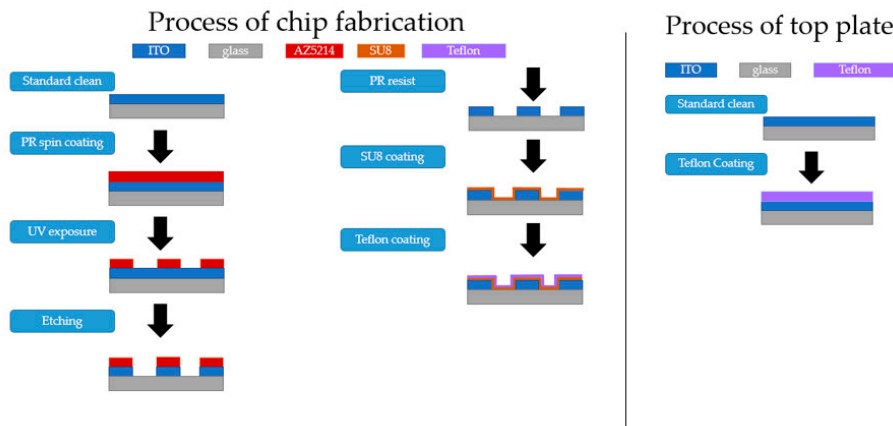

**Figure 3.** Process of EWOD chip and top plate.

### 3.2. Digital Microfluidics System

The digital microfluidic system consisted of a function generator, a power amplifier, relay boards and PXI-6512. PXI-6512 is a 64-channel, ±30 VDC industrial digital output interface, with digital I/O watchdogs which can ensure a configurable safe output state. The signal generator (33220A, Keysight, Santa Rosa, CA, USA) and the power amplifier (A304, A.A. Lab System Ltd., Ramat-Gan, Israel) mainly controlled the signal. This signal was output to the circuit through the power amplifier and connected to the relay board. The PXI-6512 (National Instruments Corp., Austin, TX, USA) was controlled with a computer program (LabView) to determine the relay switch on the relay board. The signal was accurately passed through the wires on the clamp (CCNL050-47-FRC); the clamp connected the chip and the complete system.

## 4. Results

### 4.1. Experiment with the Two-Electrode L-DEP Chip

In this experiment, propylene carbonate served as the driving liquid; the volume of the droplet was fixed at 0.33 μL. The gap was 100 μm; the applied voltages were 140 $V_{pp}$, 160 $V_{pp}$ and 200 $V_{pp}$. The frequency was 20 kHz. The chip was divided into a total width of 1 mm and an electrode of width 0.1 mm. The experiment was initiated with a single L-DEP electrode to stretch the droplet to measure the length change of the single electrode, and to turn on the two L-DEP electrodes to measure the length change of the two electrodes. Most of the droplet's volume moved to the spacing of the L-DEP electrodes when the two electrodes were activated. An example of the droplet stretching experimental images is show in Figure 4.

We found that the stretch length of the droplet decreased as the width of the electrode decreased. As the droplet completely moved to the L-DEP electrode at 200 $V_{pp}$, the change in the stretch length was small. When the chip had a fixed electrode width of 0.1 mm, the stretch length of the droplet did not vary as the electrode spacing increased, but, when the two electrodes were turned on, the stretch length gradually decreased because of the increased electrode spacing. Because the volume of the droplet was fixed, the total width of the electrode increased to cause a decreased stretch length.

In the two-electrode experiment, we found that the droplets moved completely to the L-DEP electrode; especially when the applied voltage was 200 $V_{pp}$, the amount of change in length could not be accurately measured. To solve the problem, we changed the way the droplet was manipulated. For this new manipulation, we opened the control electrode and two L-DEP electrodes when measuring the length of droplet. In this experiment, as the forces generated by the control electrode and the L-DEP electrode simultaneously affected a droplet, the stretch length was significantly decreased; most droplet volume was fixed to the control electrode.

The experimental results are shown in Figures 5 and 6. When the total width was fixed at 1 mm, the stretch length of the droplet decreased as the electrode width decreased; regardless of the magnitude of the applied voltage, it shows a downward trend. When the width of the electrode was fixed to 0.1 mm, the stretch length did not vary with the increased electrode spacing. When the two electrodes were turned on, the tendency was more obvious. Based on the phenomena above, we can use it for the comparison between experimental result and theoretical calculations. The method was applied to calculate the change in the perimeter of a droplet.

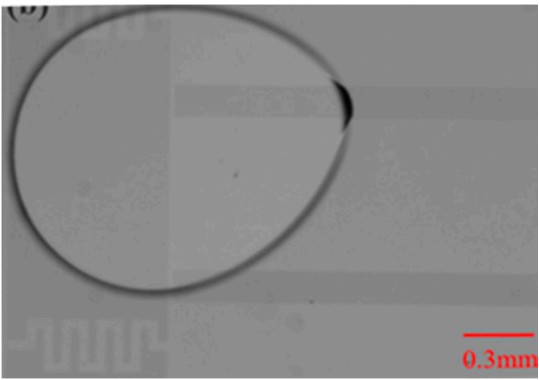

**Figure 4.** Droplet stretching experimental image.

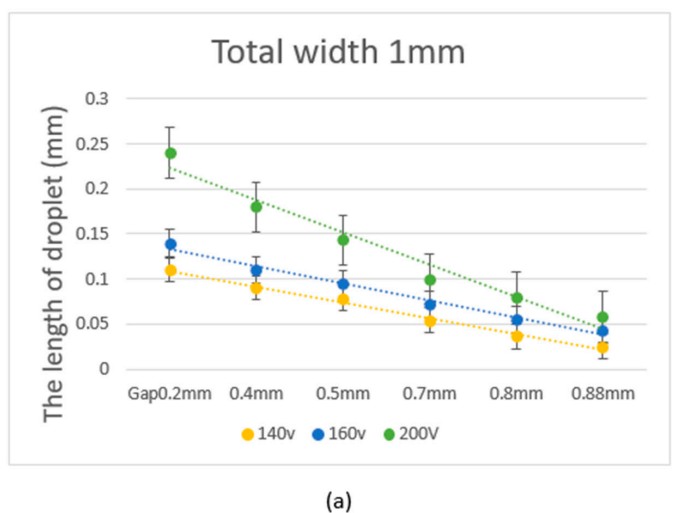

(a)

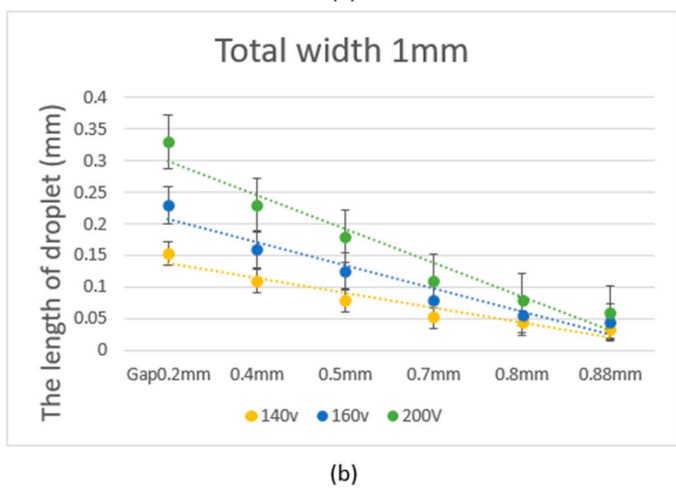

(b)

**Figure 5.** Total width fixed to 1 mm (**a**) one-electrode and (**b**) two-electrode.

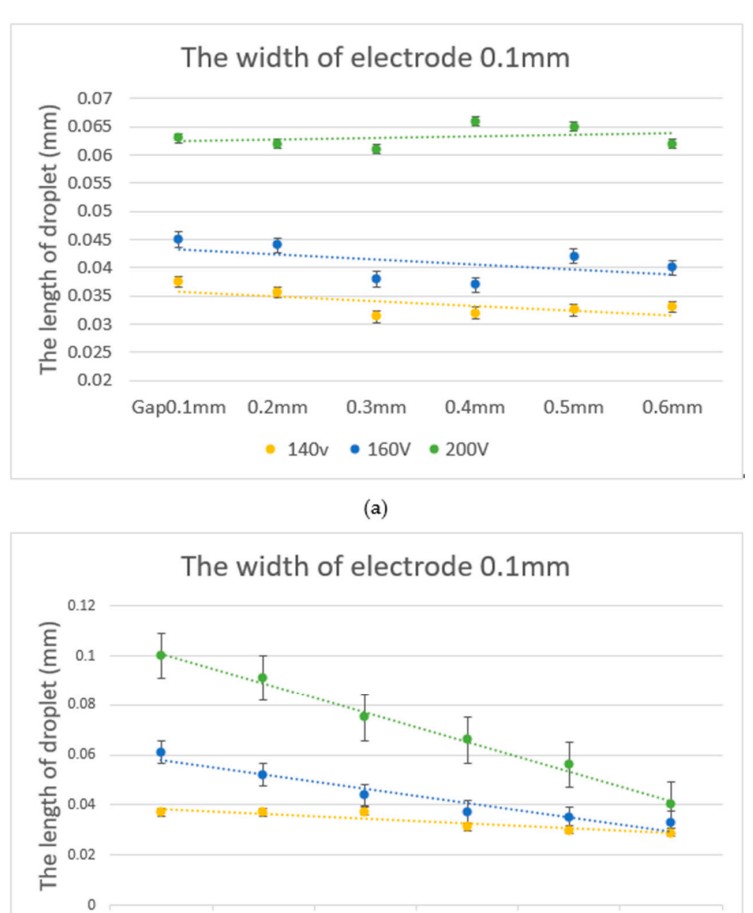

**Figure 6.** Electrode fixed to 0.1 mm (**a**) one electrode and (**b**) two-electrode.

*4.2. Comparison between Experimental Data and Theoretical Calculations*

From the above experiments, we found that the stretch length of the droplet was closely related to the electrode width, but not to the electrode spacing. We hence removed the electrode spacing terms in Equations (15) and (16). Moreover, in the case of calculating the multi-electrode case, the electrode width was multiplied by the ratio of the perimeter after the droplet was stretched, as the perimeter was related to the surface tension; the modified result is shown in Equations (18) and (19).

$$F_z^e = (w)\xi\frac{dz}{dt} + \pi(w)\gamma \tag{18}$$

$$F_z^e = \left(w \times \frac{The\ perimeter\ of\ multi-electrode}{The\ perimeter\ of\ one\ electrode}\right)\xi\frac{dz}{dt} \\ + \pi(w \times \frac{The\ perimeter\ of\ multi-electrode}{The\ perimeter\ of\ one\ electrode})\gamma \tag{19}$$

The results are shown in Figures 7 and 8. In Figure 7a,b, the experimental values exceeded the theoretical value when the voltage was applied at 200 $V_{pp}$; the other voltages were generally consistent. Figure 7a shows a theoretical value greater than the experimental value at 200 $V_{pp}$. The main error hence occurred in the case of a large applied voltage; the error might arise because the model fails to take into account the capacitance of the hydrophobic layer, resulting in a deviation between the calculated electric field force and the actual situation. Otherwise, repeated use of the test chip might result in a decreased hydrophobicity of the surface of the chip and a decreased stretch length.

To verify the correctness of the theory, we designed a three-electrode experiment to prove that this theory could be applied to single electrodes and multiple electrodes. The experimental method was the same as for the two-electrode experiment; the difference is that three L-DEP electrodes were turned on at the same time. The three-electrode chip was equally divided into an electrode of width 0.1 mm and a total width 1 mm; the experimental results are shown in Figures 9 and 10. The trends of the theoretical and experimental values are mostly the same in the two figures. To prove that this theory is applicable to multi-electrode conditions, we used the correlation coefficient to assess the correlation between the theoretical value and the experimental data, as shown in Table 1.

When the total width was fixed at 1 mm, the correlation coefficients of the two-electrode and the three-electrode chips were both near to 1, indicating almost complete correlation. When the electrode width was fixed to 0.1 mm, as the theoretical value of the single electrode does not vary with the electrode spacing to a certain value, the definition of the metric variable of the correlation coefficient was unsatisfied, thus no discussion is made. The two electrodes had a slight decrease compared with the total width, 1 mm; especially the two-electrode wafers had the greatest difference for applied voltage 140 V.

The correlation coefficients of the remaining voltages and the three-electrode wafer were, however, near to 1, which proves that this theory is applicable to multi-electrode chips.

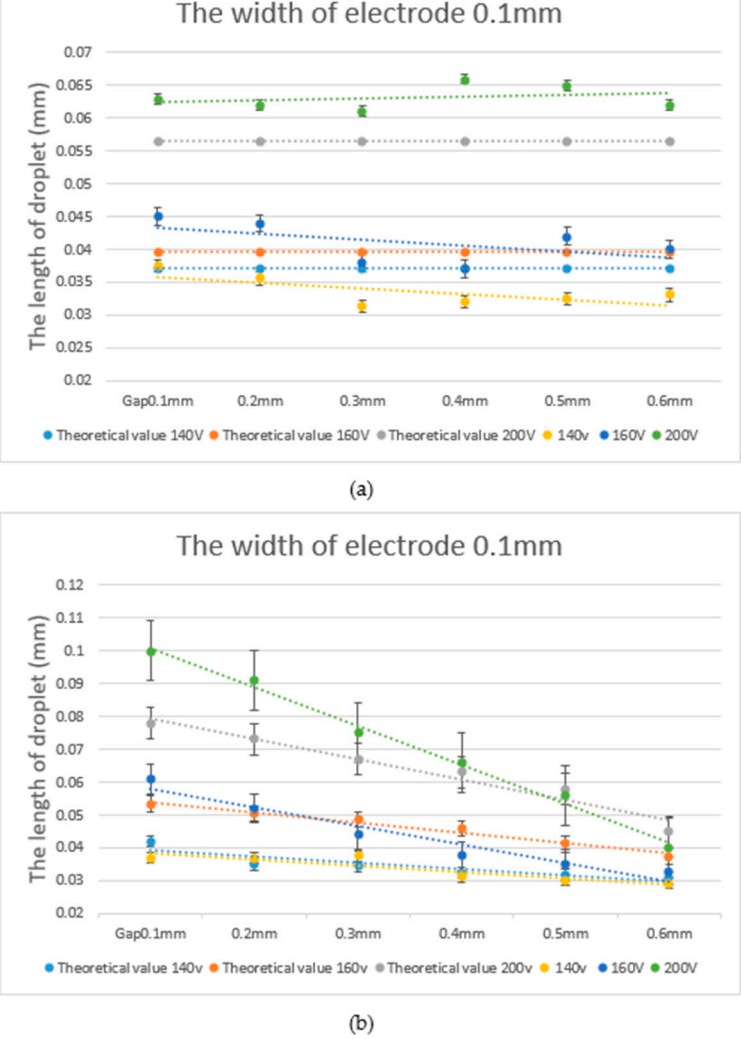

**Figure 7.** Experiment and theory for electrode width fixed to 0.1 mm (**a**) one-electrode and (**b**) two-electrode.

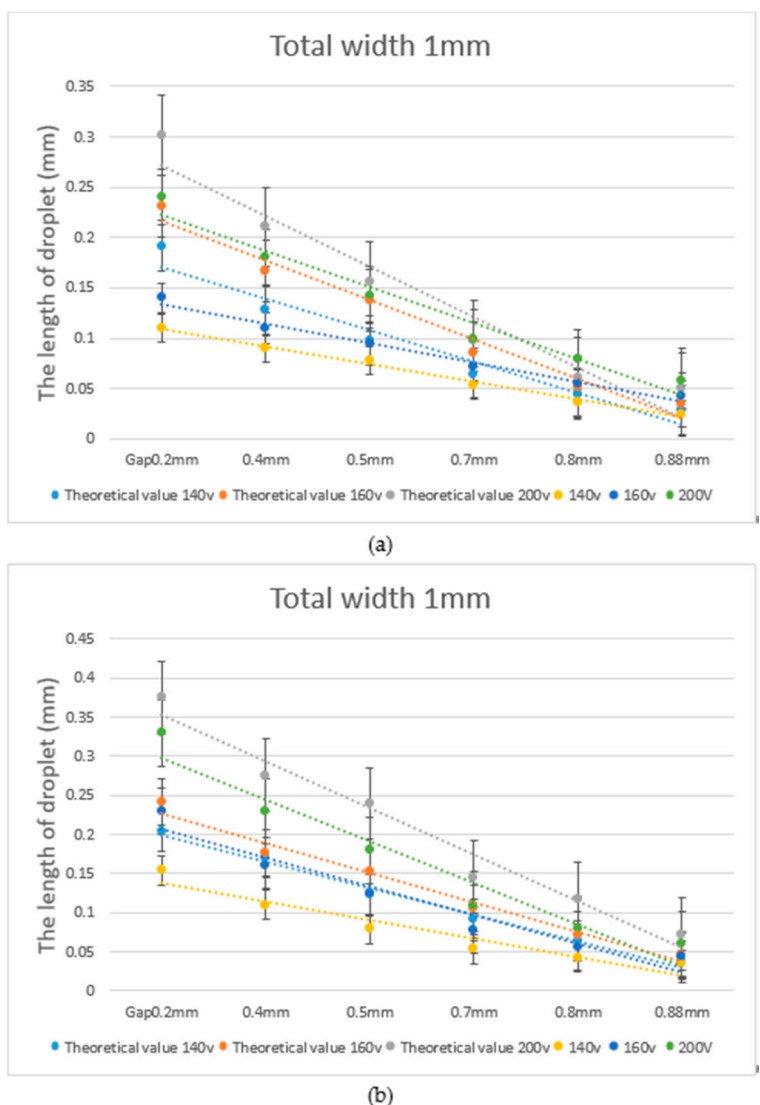

**Figure 8.** Experiment and theory for total width fixed to 1 mm (**a**) one electrode and (**b**) two-electrode.

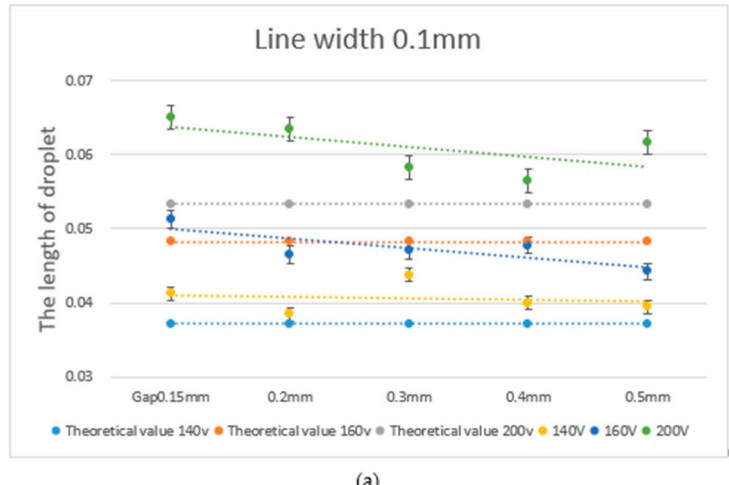

**Figure 9.** *Cont.*

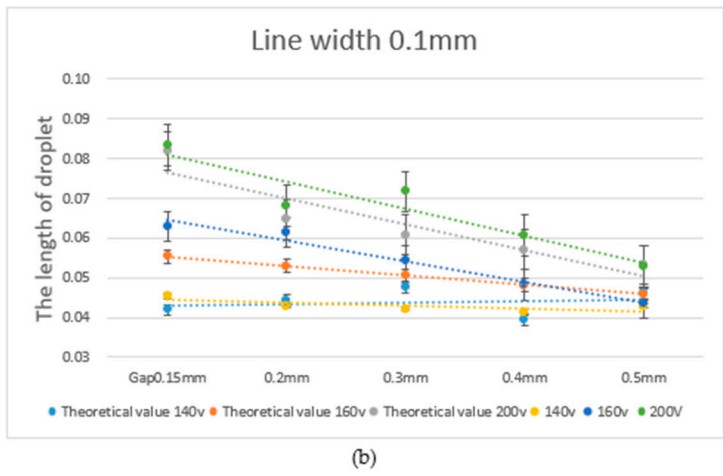

(b)

**Figure 9.** Comparison between the experiment and theory for electrode width fixed to 0.1 mm (**a**) one-electrode and (**b**) three-electrode.

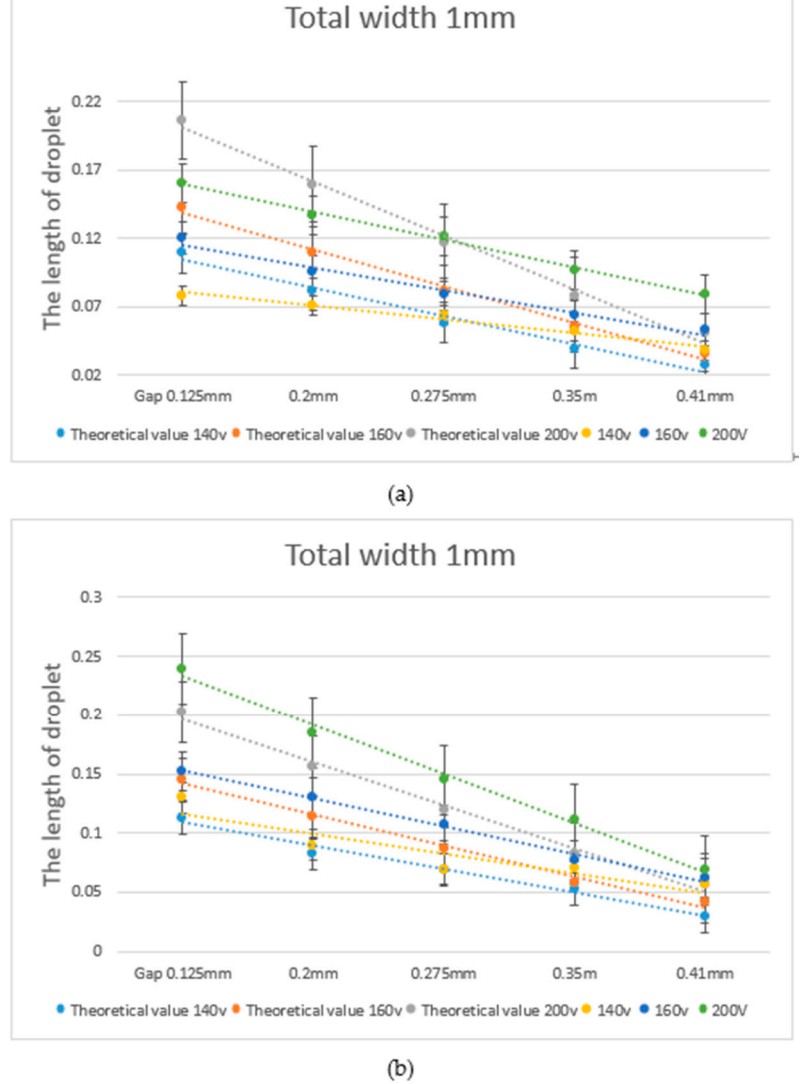

**Figure 10.** Experiment and theory for total width fixed to 1 mm; (**a**) one-electrode and (**b**) three-electrode.

**Table 1.** Correlation coefficients of the L-DEP experiment.

| Correlation coefficient | | | | |
|:---:|:---:|:---:|:---:|:---:|
| Total width 1 mm | | | | |
| | Two-electrode chip | | Three-electrode chip | |
| Applied voltage | Two-electrode | One-electrode | Three-electrode | One-electrode |
| 140 V | 0.995 | 0.978 | 0.967 | 0.948 |
| 160 V | 0.991 | 0.99 | 0.981 | 0.991 |
| 200 V | 0.994 | 0.998 | 0.986 | 0.988 |
| Electrode width 0.1 mm | | | | |
| | Two-electrode chip | | Three-electrode chip | |
| Applied voltage | Two-electrode | One-electrode | Three-electrode | One-electrode |
| 140 V | 0.717 | | 0.990 | |
| 160 V | 0.917 | | 0.986 | |
| 200 V | 0.932 | | 0.940 | |

## 5. Conclusions

In this article, we propose a theory to relate the stretch length of a droplet subjected to an L-DEP force with varied voltage, electrode width and electrode spacing, for the purpose of verifying the experiment. A propylene carbonate liquid served as a driving liquid in the experiment. The chip was divided into an electrode of width 0.1 mm and total width 1 mm. Each had six variations of the electrode spacing. The experimental results show that, as the width of the electrode decreased, the stretch length decreased, but the stretched length did not vary with an increased electrode pitch. The theory was analyzed from the point of view of the force on the droplet, but was improved based on the experimental results. To verify the theoretically improved result, we designed a three-electrode chip for the experiment. The results proved that the theory was consistent with the experimental results. The theory can hence serve to determine the stretch length of a droplet. The proposed electrode's design and distribution can be utilized in multiple applications such as patterning in accurate volumes or the amount of reagents used in biological applications. A droplet volume can be accurately controlled with a known electrode width and spacing between the upper and lower plates.

**Author Contributions:** Conceptualization, H.-T.L.; Data curation, Y.-J.C.; Formal analysis, H.-T.L.; Investigation, D.-J.Y. All authors have read and agreed to the published version of the manuscript.

**Funding:** Thanks for the funding support from Ministry of Science and Technology, Taiwan, MOST 108-2221-E-007-031-MY3.

**Conflicts of Interest:** The authors declare no conflict of interest.

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
