# Peer review of "Using a Digital Microfluidic System to Evaluate the Stretch Length of a Droplet with a L-DEP and Varied Parameters"

_inventions, doi:10.3390/inventions5020021_

Round 1

Reviewer 1 Report

In this work, the authors investigated the stretch length, thus the volume of droplet subjected to a L-DEP force in a microfluidic channel with varied voltage, electrode dimension. Although the scientific value of this manuscript is qualified for publication, the language and writing are very difficult to follow. Some English improvement is needed. Figure 2, the schematic of the chip is very hard to understand and should be improved. Figure 3, fabrication process flow is confusing too, which part is top plate? Figure caption is needed too here. Figure 4-9 has no standard deviation or error bar.

Author Response

Reviewer 1:

In this work, the authors investigated the stretch length, thus the volume of droplet subjected to a L-DEP force in a microfluidic channel with varied voltage, electrode dimension.
Although the scientific value of this manuscript is qualified for publication, the language and writing are very difficult to follow. Some English improvement is needed.

  1. Figure 2, the schematic of the chip is very hard to understand and should be improved.
    <Revision>
    Figure 2 has been improved. Because we need to generate the droplet from reservoir, we need to design the reservoir electrodes with one pairs of control electrode inside, shown in the left of figure 2a. Then the design of control electrodes was shown in the right of figure 2a. In order to let reader understand “two electrode” and “three electrode” systems, we tried to show the design of
    those two systems with fixed total width and fixed line, shown in figure 2b and 2c.
  2. Figure 3, fabrication process flow is confusing too, which part is top plate?
    Figure caption is needed too here.
    <Revision>
    Figure 3 has been improved.
  3. Figure 4-9 has no standard deviation or error bar.
    <Revision>
    The standard deviation has been shown from figure 5 until figure10.

Reviewer 2 Report

The authors present analytical modeling to support the experimental designs for liquid dielectrodphoresis. The correlation between experimental results and theoretical modeling is important to consider experimental parameters. Although the authors report  the parameters that support the modeling, the manuscript requires to modified and improved to be published to the Journal of Inventions. My comments and suggestions are listed below.

The author should add theses references, “Droplet creation using liquid dielectrophoresis”, Chen et al, 2009, Sensors and Actuators B: Chemical and “A liquid-metal-based dielectrophoretic microdroplet generator”, Wang et al., 2019, Micromachines

Line 88: Figure 1(a) schematic diagram doesn’t seem to match with Figure 3 cross-sectional view. At the bottom layer, there is Teflon coating and there is no SU-8 at the top layer.

Line 64 (158): What is the “bead” here? Droplet?

Line 147: There should be a space between “was” and “100”.

Line 161-162: The sentence is not clear.

Line 172: … experimental results to discuss with the theory” This sentence is not clear.

Line 221: What about the effects of frequency? It would be appreciated if authors do more studies varying frequencies.

Figure 4-9, all the images quality should be improved.

It is not clear how the droplet is stretching, the authors should show an example of experimental images in the Results section.

The authors proposed experimental results to support the analytical modeling. The initial discrepancy of the experimental results and the analytical approach would be important to optimize the parameters (electrode gap, distance and others). I would suggest the authors to add analytical calculations in Figure 4 and 5. If one of the electrodes provides a better prediction, then only this parameter is considered the rest of the study.

 I would recommend the authors comment on this L-DEP approach for practical applications in the Conclusions section

Round 2

Reviewer 2 Report

One of the main questions about this manuscript is droplet volume vs. analytical variation (experiment). The authors claims that “known volume” is the key parameter for L-DEP dispensing and manipulation. How this volume is applied to the model and how is different when the experiments are performed in different volumes?

Line 25: The sentence is clear.

Line 33: I would correct “equipment” to “system”.

Line 38: “Droplet microfluidics” also should be included.

Line 40: The sentence is not clear.

Line 45: In terms of fabrication for the devices, I would not say it’s low cost.

Line 148: The fixed droplet volume is 0.33 uL. The authors requires to explain how this volume is dispensed on the substrate.

Line 160: “bead” should be “droplet”.

Line 164: This sentence is not clear.

Line 175: If the “velocity” term is applied, it should also mentioned in the Results/Discussions section.

Line 240: “etc.” should be removed.

Author Response

Reviewer 2:

One of the main questions about this manuscript is droplet volume vs. analytical variation (experiment). The authors claims that “known volume” is the key parameter for L-DEP dispensing and manipulation. How this volume is applied to the model and how is different when the experiments are performed in different volumes?

 <revision>

The known volume was defined by the design electrode size and gap between chip and top plate. Liquid is placed on L-DEP chip’s reservoir and then cover top plate, to let the liquid manupulate in the gap. Droplet is generated by control electrodes from reservoir electrodes. We did all the experiments under fixed volume.

Line 25: The sentence is clear.

Thanks for the suggestion.

“The theory was based on the force balance on a droplet, that involved the force generated by the electric field, friction force, and capillary force. “

Line 33: I would correct “equipment” to “system”.

Thanks for the suggestion. The “equipment” has been revised to be “system”.

Line 38: “Droplet microfluidics” also should be included.

Thanks for the suggestion. The “droplet microfluidics” has been added.

Line 40: The sentence is not clear.

Thanks for the suggestion. The sentence is change to “The lab-on-a-chip (LOC) technique uses a digital microfluidic system to transport and to manipulate a small amount of liquid”

Line 45: In terms of fabrication for the devices, I would not say it’s low cost.

<revision>

Base on EWOD and L-DEP chip’s high reliability and reusable features, so here said it’s low cost. Also, compare with PDMS which need to make the film out of mother mode, it may cost damage to mother mode. It relatively expensive when remaking a mode.

Line 148: The fixed droplet volume is 0.33 uL. The authors requires to explain how this volume is dispensed on the substrate.

<revision>

Liquid is placed on L-DEP chip’s reservoir and then cover top plate, to let the liquid transfer in the gap. Droplet is generated by control electrodes from reservoir electrodes.

Line 160: “bead” should be “droplet”.

Thanks for the suggestion. The “bead” has been revised to be “droplet”.

Line 164: This sentence is not clear.

<revision>

For new manipulation, we open control electrode and two L-DEP electrodes when measuring the length of droplet.

Line 175: If the “velocity” term is applied, it should also mentioned in the Results/Discussions section.

<revision>

Thanks for the suggestion. Sentence has been revised to be” The method was applied to calculate the change in the perimeter of a droplet.”

Line 240: “etc.” should be removed.

Thanks for the suggestion. “etc.” is removed.